# Optimization of Protocol for Construction of Fungal ITS Amplicon Library for High-Throughput Illumina Sequencing to Study the Mycobiome of Aspen Leaves

Abu Bakar Siddique [1], Benedicte Riber Albrectsen [1], Hulya Ilbi [2] and Abu Bakar Siddique [3],*

1   Umeå Plant Science Centre (UPSC), Department of Plant Physiology, Umeå University, 90187 Umeå, Sweden; abu.bakar.siddique@umu.se (A.B.S.); benedicte.albrectsen@umu.se (B.R.A.)
2   Department of Horticulture, Faculty of Agriculture, Ege University, Izmir 35100, Turkey; hulya.ilbi@ege.edu.tr
3   Department of Ecology and Environmental Sciences, Umeå University, 90187 Umeå, Sweden
*   Correspondence: abu.siddique@umu.se

**Abstract:** High-Throughput Illumina Sequencing (HTS) can be used to study metagenomes, for example, those of importance for plant health. However, protocols must be optimized according to the plant system in question, the focal microorganisms in the samples, the marker genes selected, and the number of environmental samples. We optimized the protocol for metagenomic studies of aspen leaves, originating from varied genotypes sampled across the growing season, and consequently varying in phenolic composition and in the abundance of endo- and epiphytic fungal species. We optimized the DNA extraction protocol by comparing commercial kits and evaluating five fungal ribosomal specific primers (Ps) alone, and with extended primers that allow binding to sample-specific index primers, and we then optimized the amplification with these composite Ps for 380 samples. The fungal DNA concentration in the samples varied from 561 ng/μL to 1526 ng/μL depending on the DNA extraction kit used. However, binding to phenolic compounds affected DNA quality as assessed by Nanodrop measurements (0.63–2.04 and 0.26–2.00 absorbance ratios for 260/280 and 260/230, respectively), and this was judged to be more important in making our choice of DNA extraction kit. We initially modified the PCR conditions after determining the concentration of DNA extract in a few subsamples and then evaluated and optimized the annealing temperature, duration, and number of cycles to obtain the required amplification and PCR product bands. For three specific Ps, the extended Ps produced dimers and unexpected amplicon fragments due to nonspecific binding. However, we found that the specific Ps that targeted the ITS2 region of fungal rDNA successfully amplified this region for every sample (with and without the extension PP) resulting in the desired PCR bands, and also allowing the addition of sample-specific index primers, findings which were successfully verified in a second PCR. The optimized protocol allowed us to successfully prepare an amplicon library in order to subject the intended 380 environmental samples to HTS.

**Keywords:** NGS; metabarcoding; metagenomics; eDNA; ITS; endophytes; amplicon; aspen; rDNA

## 1. Introduction

It is increasingly being recognized that microorganisms may determine the functioning and health of eukaryotes, whether plants, humans, or other animals. There is therefore great scientific interest in studies of microbial communities (or metagenomes) in environmental samples (a term which, in addition to biological tissues, also refers to samples of surrounding substrates such as air, water, or soil [1,2]). High-Throughput Illumina Sequencing (HTS), which allows for amplicon sequencing at great depth and high taxonomic resolution, is becoming a preferred method for studying metagenomes in large sets of environmental samples. As a cultivation-independent method it is frequently used to study the diversity and potential function of microbial communities [1–3]. Mycobiomes (or microbial fungi) in the phyllosphere of trees have been studied to investigate their

association with tree performance related to plant internal and external factors such as plant genotype [4,5], organ specificity [6], and geographic and climatic parameters [7–12]. Sequencing protocols and bioinformatic pipelines that are used to dissect metagenome data are also under rapid development [1,3]. As environmental samples vary from project to project the molecular workflow must be optimized in each case to target the right organisms in an appropriate way [3,13–16]. Firstly, DNA extraction kits used for metagenomic studies must operate within the physicochemical constitution of the environmental sample in question [3]. Secondly, sample primers must be designed to bind only to the microorganisms of interest, and they should operate reliably across samples that may vary in, for example, DNA content [3]. Thirdly, as single samples for HTS are labelled and pooled during the sequencing process, an extension and index primer system must be designed to construct an adequate amplicon library, and this affects the design of the primers and the PCR settings for amplification of the DNA extracts [16]. Finally, the entire protocol must be optimized to allow for comparisons of metagenomic data [17]. Here, we present the optimization of a HTS protocol for fungal metagenomic studies of large set of aspen (*Populus tremula* L.) leaves (Figure 1a) that varied in genotype, age, phenolic composition and the expected richness and abundance of endo- and epiphytic fungal species. During the optimization, we evaluated five fungal ribosomal DNA-specific primers to extract the mycobiome, and developed and tested four extended primers as well as two index primers to identify the mycobiome of 380 samples. Before preparing all samples for HTS, a subset of samples was selected for each of the optimization steps, from DNA extraction to specific polymerase chain reaction (PCR) protocols, which were optimized for fungal, extension and index primers.

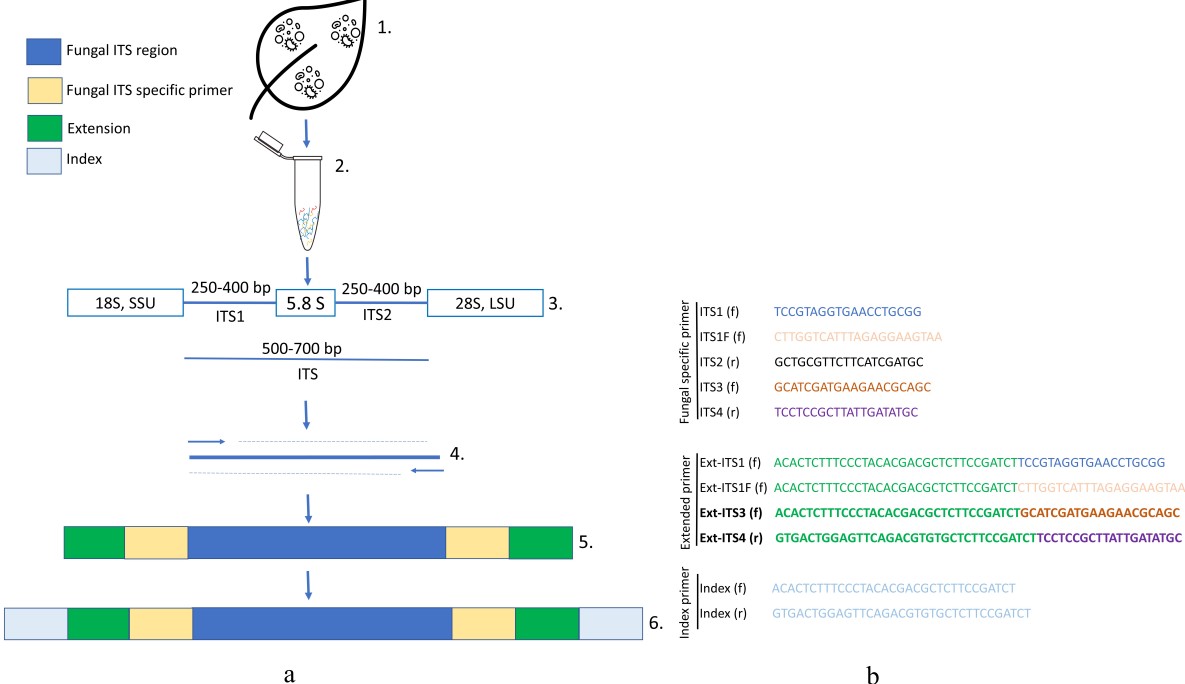

**Figure 1.** Workflow for preparing an amplicon library for high-throughput sequencing (HTS): (**a**) (1) leaves were harvested, lyophilized, and ground to powder; (2) DNA was extracted; (3) primers were designed to target the conserved fungal rDNA ITS region; (4) forward and reverse fungal-specific primers were generated for PCR; (5) extended PCR products were made, containing the ITS region (blue), the fungal-specific primer region (yellow), nucleotides as an overhang (green) and binding site for index primers; (6) addition of sample-specific index primers (pale blue) to the extension PCR product from (5), which were then compiled to create the amplicon library. (**b**) Primers tested in this study (selected primers used in bold).

## 2. Materials and Methods

### 2.1. Biological Materials

Environmental samples for this project consisted of aspen leaves from the TanAsp common garden experiment (Vindeln, Sweden) [18]. In short, during 2020 and 2021, at monthly sampling events (July to September) ten mature undamaged leaves were randomly harvested from the canopies of 72 aspen trees (*Populus tremula*). The trees had been planted in 2010 and were up to 510 and 550 cm high in June 2020 and 2021, respectively, and sampling could in most cases be performed at breast height. During harvest, leaves were cut at the base, leaving the petiole mostly undamaged, after fluorescence measurement to assess pigments (DUALEX® Optical leafclip meter, Force-A, Orsay, France); leaf samples consisting of all leaves per tree were placed in labeled plastic bags, and flash frozen in the field on dry ice, then transported ca 1 h (hour) to the lab in a cooler (Adriatic 24 L). Bags were stored at −80 °C in freezers at UPSC prior to lyophilization for 24 h in the freeze dryer (LABOGENE; 3450 Lillerød, Denmark). The dried leaves were then ground using a mortar and pestle to create a fine ($10^{-3}$–$10^{-2}$ cm) powder for DNA extraction. Ca 750 mg powder was obtained per sample, corresponding to more than 15 times the amount needed for the DNA extraction procedure. All handling of the leaves was performed with sterile gloves (Nitrile Ambidextrous Gloves, ThermoFisher Scientific, Göteborg, Sweden) to prevent contamination with foreign DNA, and containers were sterilized between sample handlings.

### 2.2. DNA Extraction

Two extraction kits were compared for use with aspen leaves: E.Z.N.A. Plant DNA Kit (Omega Bio-tek Inc., PW, Norcross, GA, USA) and ChargeSwitch gDNA Plant Kit (ThermoFisher Scientific, Göteborg, Sweden). In both cases, we followed the provider's protocol and used 50 mg of leaf powder (corresponding to ca 1/15 of the material, ~half a leaf). DNA concentrations were measured using a Nanodrop (ND-1000 spectrometer) to determine absorbance ratios: 260/280 and 260/230 (Table 1).

**Table 1.** Comparison of DNA extraction kits using Nanodrop measurements for assessment. Extracted DNA (1 µL) from two samples was measured with an ND-1000 spectrometer. Sample ID refers to identity of samples from the TanAsp collection, 260/280 is a measurement of DNA/protein ratio, and 260/230 is a measure of DNA/contaminant ratio. The preferred kit was chosen on the basis of both better ratios.

| Kit | Sample ID | DNA Concentration (ng/µL) | 260/280 | 260/230 |
|---|---|---|---|---|
| E.Z.N.A. Plant | 253 | 561.1 | 2.04 | 1.99 |
| DNA | 281 | 804.3 | 2.02 | 2.00 |
| ChargeSwitch | 253 | 1526.3 | 0.63 | 0.29 |
| gDNA | 281 | 1311.8 | 0.63 | 0.26 |

Expected 260/280 value is~1.8 and it is generally accepted as "pure" for DNA. Expected 260/230 values are commonly in range of 2.0–2.2.

### 2.3. Primers, PCRs, and Gel Electrophoresis

Primers were designed according to recommendations for fungal metabarcoding [19] that, together with sample-specific index primers (with added adapters and overhangs as suggested by SciLifeLab National Genomics Infrastructure, NGI), enabled us to complete the design of extended primers for our sample library (Figure 1b). In total, we used five fungal-specific primers, four extended primers, and two primers for attachment to sample-specific index primers (Figure 1b).

In general, 25µL PCR mixture was used as consisting of 1X Dream Taq buffer, 0.16 µM dNTP mix, 0.4 µM forward and reverse primers, ~0.25 µg template DNA and 0.75 unit Dream Taq DNA polymerase and PCR grade water. Each PCR run was initiated with the following settings: initial denaturation at 95 °C for 3 min followed by 30 cycles of 95 °C for

30 s (denaturing), 56 °C for 60 s (annealing), 72 °C for 60 s (elongation) and final extension at 72 °C for 7 min. To judge the quality of a PCR run the products were visualized under UV light after gel electrophoresis in a 1% agarose gel containing GelRed® (Biotium, Inc., Fremont, CA, USA) (3 μL/100 mL agarose gel) at 140 volts for 30 min.

　　　The five fungal-specific primers (Figure 1a, point 4) were tested according to the initial protocol settings above and optimization adjustments were performed mainly by the choice of fungal ribosomal primers specific to the ITS region they targeted.

　　　The extended PCRs step tested the binding of the extension (Figure 1a, point 5). Four extension primers were designed to test how well the extension adhered to the fungal primer pairs (Figure 1a, point 4). We did not include any primer pairs targeted for the ITS1 region as their respective fungal-specific primer pairs failed to amplify the ITS1 region. We included extended primer pairs responsible for the entire ITS and for the subregion ITS2 (Figure 2). To optimize for a lower amount of template DNA, forward and reverse primers (0.5 μL) were used when no bands were found. The annealing temperature was adjusted back and forth from the calculated temperatures to obtain the best bands and cycle number was increased to increase band intensity during PCR improvement (for details and values, please refer to Table 2 and Figure 2).

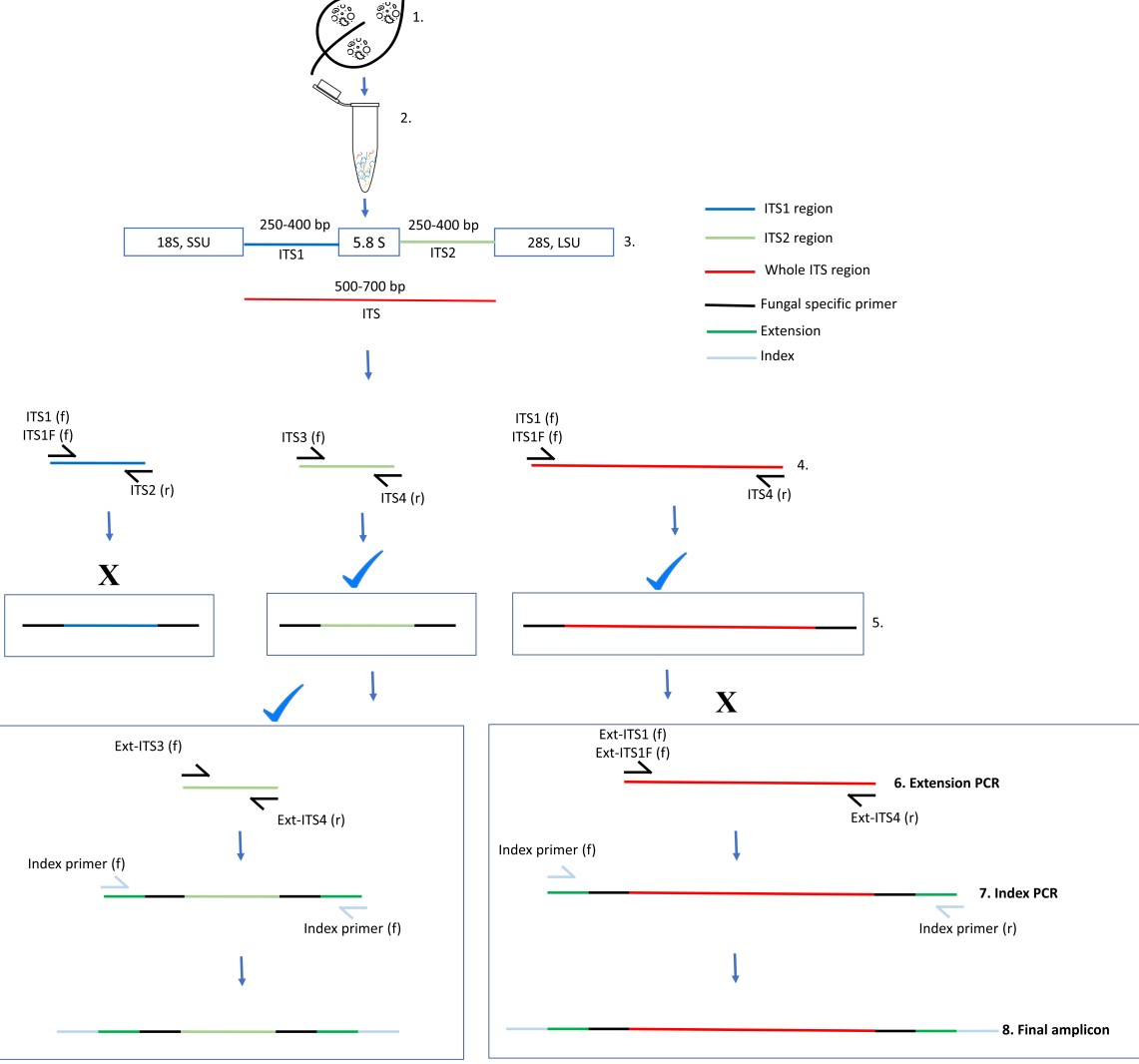

**Figure 2.** Schematic diagram showing the sequences of primers and general sequential steps for fungal amplicon library preparation. (1) HTS (Illumina MiSeq) sequencing can detect genetic material from many kinds of organisms found in an environmental sample. We were studying the leaves of

aspen and were interested in detecting the composition of the leaves' fungal microbiome. (2) After freeze drying and grinding leaves, the DNA was extracted using kits and its quality and quantity were measured to check the purity. (3) Fungal DNA can be distinguished from DNA from other organisms through specific and highly conserved regions in the ribosomal genes referred to as internal transcribed spacer (ITS) regions, which are not shared by plants or bacteria. rDNA consists of five regions in repeated gene clusters: the highly conserved ITS1 and ITS2 are separated by three coding regions: the central 5.8S subunit, the 18S and the 26S unit at each end of the rDNA cluster 400–900 base pairs long. (4,5) Fungal-specific primer pairs are used to detect and amplify the specific ITS region. In our case, fungal-specific primer pairs successfully amplified the specific region containing the whole ITS and ITS2, whereas ITS1 (f) + ITS2 (r) failed to amplify the ITS1 region. Based on this, we did not include the extended primer for the ITS1 region. (6) Extended primer pairs were used to amplify the ITS and ITS2 regions of fungal rDNA. (7,8) Extended primer pairs that targeted the whole ITS region failed to amplify at different annealing temperatures and PCR cycles. As a result, index PCR was unsuccessful. However, extended primer pairs successfully amplified the ITS2 region and attached the index primer to the index PCR.

**Table 2.** An overview of the optimization procedure for preparation of aspen leaf metagenomics samples for HTS sequencing. Letters (a–i) refer to the resulting PCR gels as shown in Figure 3. Template DNA refer to volume of sample DNA extract. Settings = PCR settings indicated as "annealing temperature °C"/"number of cycles".

| Purpose | Primers | Template DNA | Settings | Outcome | Decision/Selection |
|---|---|---|---|---|---|
| Fungal PCR for ITS-selection | ITS1 (f) + ITS4 (r) | 1. 1 µL | 1. 56/30 * | 1. SB with desired DF (a) | 1. Pp failed to bind to fungal ITS region. |
| | ITS1F (f) + ITS4 (r) | 1. 1 µL | 1. 56/30 2. 57/30 | 1. NB 2. SB with desired DF (b) | 1. Pp failed to bind to ITS. 2. Sucessful amplification of ITS. |
| | ITS1 (f) + ITS2 (r) | 1. 1 µL | 1. 56/30 2. 57/30 | 1. NB 2. NB | 1. Pp failed to bind to ITS1. 2. Pp failed to bind to ITS1. |
| | ITS3 (f) + ITS4 (r) | 1. 1 µL | 1. 57/30 | 1. SB with desired DF (c) | 1. Pp successfully amplified ITS2. |
| Ext. PCR for ITS ampl. | Ext-ITS1F (f) + Ext-ITS4 (r) | 1. 1 µL 2. 1 µL 3. 1 µL 4. 1 and 0.5 µL 5. 1 and 0.5 µL 6. 1 and 0.5 µL 7. 1 and 0.5 µL 8. 0.5 and 1 µL PCR Product 9. 1 and 0.5 µL | 1. 62/30 2. 60/30 3. 58/30 4. 62/30 5. 64/30 6. 66/30 7. 68/30 8. 60/30 9. 58/33 | 1. FB (d) 2. FB with PD 3. FB with undesired DF (e) 4. FB with undesired DF 5. FB with undesired DF 6. FB with undesired DF 7. FB with undesired DF 8. SB with desired DF 9. PD with both desired and undesired DF (f) | 1–7. Ext. Pp failed to amplify the ITS. 8. Ext. Pp successfully amplified ITS 9. Ext. Pp amplified successfully. |
| | Ext-ITS1 (f) + Ext-ITS4 (r) | 1. 1 µL 2. 1 µL | 1. 56/30 2. 58/30 | 1. NB 2. NB | 1. Ext. Pp failed to amplify ITS. 2. Ext. Pp failed to amplify ITS. |
| Index PCR | Index primer pair | 1. 8 µL of PCR product of Ext-ITS1F (f)+ Ext-ITS4 (r) | 1. 55/8 | 1. FB with PD (h) | 1. Index Pp failed to amplify, suggesting absence of ext. region in PCR products. |
| Ext. PCR | Ext-ITS3 (f) + Ext-ITS4 (r) | 1. 1 µL 2. 1 µL 3. 1 µL | 1. 72/33 2. 70/33 3. 57/33 | 1. NB (g) 2. FB (g) 3. SB with desired DF (g) | 1. Ext Pp failed to amplify ITS2. 2. Ext Pp failed to amplify ITS2. 3. Ext Pp amplified ITS2. |
| Index PCR for ITS2 ampl. | Index primers | 1. 8 µL PCR product of Ext-ITS3 (f) + Ext-ITS4 (r) | 1. 55/8 | 1. Desired difference between index PCR and Ext. PCR(i). | 1.Index Pp successfully amplified extended overhang with index for amplicon library. |

* 1. 56/30 means '1' is the serial number of the repeated step, '56' is the annealing temperature and '30' is the number of PCR cycles. Abbreviations: DF = DNA Fragment; FB = faint band; Ext. = extended; NB = no band; SB = strong band; PD = primer dimers, Pp = primer pairs; ampl = amplification.

To add index primers to the samples, commercial index primers that could bind uniquely to the samples were provided by SciLifeLab (https://ngisweden.scilifelab.se/methods/illumina-amplicon-sequencing, access date 5 December 2021). Index PCR was performed on five randomly selected extended PCR products to confirm the attachment of

index primers (Figure 1a, point 6). The same protocol was followed as described above for fungal PCR, although extended PCR products replaced template DNA, and index primers replaced forward and reverse primers. Following the SciLife protocol, up to 8 μL extended PCR product (equivalent to 5 ng of sample) was added to provide a total volume of 20 μL. The PCR protocol followed SciLifelab's recommendations: initial denaturation at 98 °C for 2 min followed by 8 cycles of 98 °C for 20 s (denaturing), 55 °C for 20 s (annealing), 72 °C for 15 s (elongation) and final extension at 72 °C for 2 min.

For each PCR we randomly selected four to ten subsamples from the 380 environmental samples to optimize the protocols for preparing the Illumina amplicon library. The final settings were as mentioned above with similar PCR settings (except for changes to the extension PCR: 57 °C as annealing temperature and 30 PCR cycles) targeting the ITS2 region and were applied to all components of the amplicon library that was sent for Illumina sequencing at NGI SciLifeLab, which carries out clean-up with MagSI beads (following SciLifeLab's protocol) to eliminate primer-dimers and undesired amplicons followed by multiplexing with 380 UniQue Dual Indexes (UDI), before sequencing using an Illumina MiSeq v3 2 × 300 bp (paired-end) in NGI SciLifeLab.

## 3. Results and Discussions

### 3.1. Choice of Extraction Kit for Fungal DNA Extraction

Generally, to test the efficiency of a DNA extraction, DNA concentration and purity were assessed. For DNA extraction, we used an E.Z.N.A. Plant DNA Kit, which yielded a lower DNA concentration but gave higher absorbance readings at 260/280 and 260/230 compared to the ChargeSwitch gDNA kit (Table 1). The higher 260/230 and 260/280 values indicate a greater purity and thus better quality [20,21]. The lower 260/230 values could be caused by the presence of phenolic compounds, salts, and/or solvents in the samples [22,23]. As aspen leaves are rich in phenolic compounds and as the presence of phenolics and proteins can hamper the downstream amplicon library preparation [24], the ChargeSwitch DNA Kit was judged to be suboptimal for aspen leaves if the regular washing steps were followed. Psifidi et al. [22] added an extra washing step. Given our large number of samples, we valued the time that could be saved by omitting additional washing steps, and instead we opted for the E.Z.N.A. Plant DNA kit, which was also superior in terms of the quality of the DNA that it generated.

One of the most important steps when extracting fungal DNA involves the removal of the thick layer of polysaccharides, proteins, and glycoproteins, melanin, chitin, and other polymers that encapsulate the fungal mycelium and provide an enzymatic and chemical barrier [25,26]. Both of the DNA extraction kits that we selected bind and remove cellular contaminants. The E.Z.N.A. kit uses a spin-based method to absorb the DNA in a spin column before it is released by elution. ChargeSwitch uses an approach in which the DNA binds to a magnetic solid surface coated with binding antibodies, after which contaminants can be removed by increasing the number of washing steps. We used Nanodrop readings to assess the quality of the extracted DNA in our samples. The advantage of this method is that it allows for the processing of many samples. Another possibility is to employ a fluorometric method such as Qubit which may strengthen the decision of extraction kits, which may provide stronger support for decisions regarding choice of extraction kit and protocol optimization. We only tested the DNA extraction protocol on two samples; more robust testing could be achieved by increasing the numbers of both biological and technical replicates.

### 3.2. Optimization of Fungal Primer Pairs

Of the five fungal ribosomal-specific primer pairs designed for this study (Figure 1b), two pairs amplified fungal ITS1, one pair ITS2, and two pairs the entire fungal ITS region (Table 2). The ITS unit is 500–700 bp long with two subunits (ITS1 and ITS2) of 250–400 bp each, separated by the 5.8S unit. To prepare the amplicon library for HTS, any one of the three ITS region may be targeted, and we initially tested all three options and found that all

ITS primer pairs resulted in the best, strongest, clearest outcome at this step except for the primer pair targeting the ITS1 region (Figure 2).

The PCR outcomes guided us in making a decision about primer choice (Table 2). The band position should match the expected number of base pairs (bp; Figure 3a–c,f,i); an absence of band indicated that the desired ITS-DNA sequence had not been amplified (Figure 3e,f), while undesired band positions (Figure 3f,h) suggested primer mismatches or primer-dimers, which are considered to result from hybridization between two primers (Figure 3d,e). Faint bands were interpreted as inefficient amplification and were adjustments were made by changing the annealing temperature (Figure 3g) and in some cases by changing the number of cycles during amplification (Figure 3f,g).

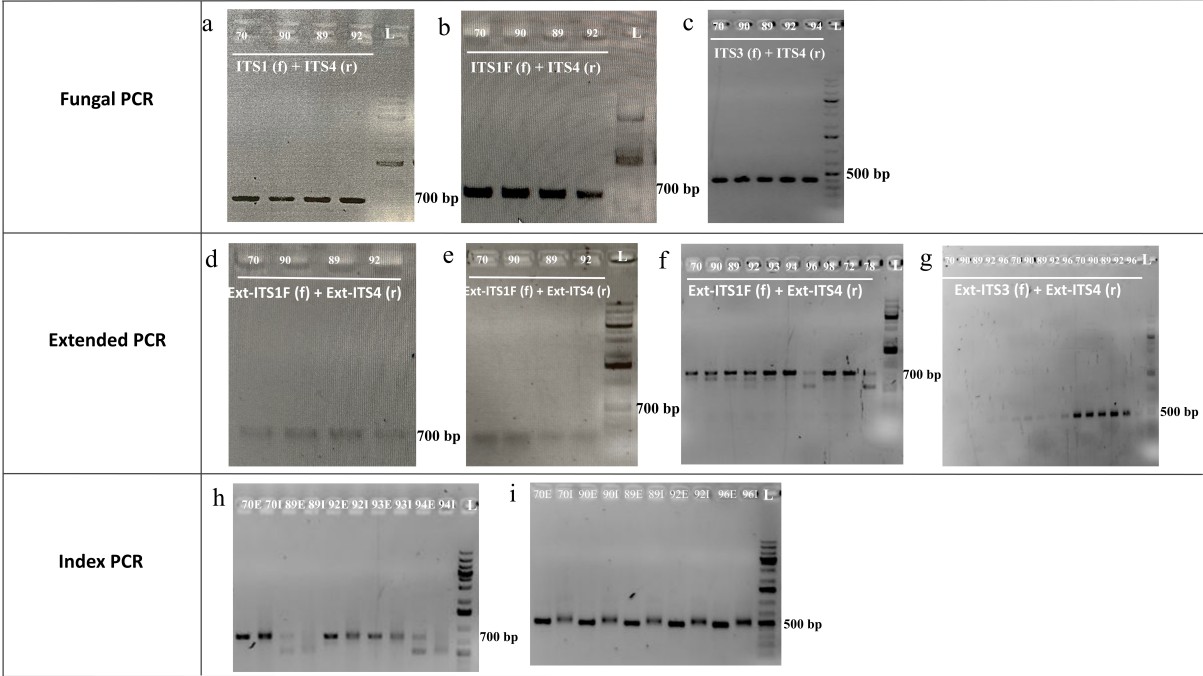

**Figure 3.** Gel images illustrating PCR optimization steps (Table 2), including test of primer function related to Figure 1. Numbers in the gell's wells refer to sample numbers from the aspen TanAsp-collection [18]. L = Ladder, indicates fragment size in number of base pairs (bp). Desired fragments of length of 500–700 bp were found for (**a,b**), and fragments of length of 250–400 bp were found for (**c**). Weak bands were obtained for (**d**) and (**g**, first nine wells). Dimers (the result of primer hybridization) were found for (**d,e**). Undesired fragments of length of 400–500 bp were found for (**f,h**). Desired fragments from extended PCR were found for (**g**, last five dark bands). Desired fragments (**g**) and index PCR fragments were compared with (**i**).

In PCR, the formation of primer dimers is one of the most common problems, which it has been suggested is caused by hybridization within and between primer pairs [27,28]. Many solutions for avoiding them are suggested in the literature, including: careful primer design, hot start PCR, touch-down PCR, changing the amount and concentration of primers and template DNA, and changing the annealing temperature, duration, and number of PCR cycles [29–35]. We changed the annealing temperature, duration, number of cycles and amount of template DNA, resulting in a single band without primer-dimers from each reaction (for detailed settings, please see Table 2).

### 3.3. Optimization of Extension PCR

With the initial PCR settings, extended primers, designed for long reads (ITS), resulted in faint or no bands (Table 2; Figure 3d,e). However, the bands became stronger at lower annealing temperature (58–60 °C) and with more PCR cycles (33, Figure 3f,g). Adjustments to annealing temperature, PCR cycles, and DNA concentration (Table 2, Figure 3f) were used

to remove dimers of long (extended) primers, which are known to produce primer dimers or no bands [36,37]; this may be solved with a two-step PCR [38]. Using two-step nested PCR, we successfully amplified the first PCR product (not shown), but the second step, during which we increased the number of PCR cycles to obtain more of the PCR products, failed to amplify. The extended primer pairs that failed to work were discarded and instead we used extended primers binding to the ITS2 subregion, which is also used as a marker for mycobiome diversity analyses [39]. We focused on amplicon library preparation based on the ITS2 region of fungal rDNA using extended primers three and four (Figures 1b and 3g) at different annealing temperatures (Table 2). These primer pairs successfully amplified the ITS2 region at 57 °C with 33 cycles along with the attachment of extended sequence parts (Table 2 and Figure 3g). We started the optimization for the ITS region with a lower number of PCR cycles, but this resulted in no bands. Based on this observation, we started the optimization of ITS2 region amplification by using a larger number of PCR cycles. A large number of PCR cycles might reduce the risk of intra- and inter-attachment of primer pairs and so might result in single bands without the primer dimers [36]. Our extended primer pairs with more PCR cycles also amplified the ITS2 regions giving very strong bands in a gel (Table 2, Figure 3g). So, for the remainder of our 380 samples, we used 30 cycles as additional PCR cycles may risk including the amplification of uncommon sequences, producing chimeric sequences [3]. Thus, we accomplished the optimization of PCR set-up for extended primers as follows: initial denaturation (94 °C for 5 min), 30 cycles of denaturation (94 °C for 1 min), annealing (57°C for 1 min) and extension (72 °C for 30 s) with a final extension (72 °C for 2 min). Modification of annealing temperatures and cycle number along with the amount of template DNA were the core of the strategy that we followed to ascertain the best combinations of PCR reactions and PCR set-up, findings that may help people to achieve PCR and primer optimization. However, other elements in PCR reaction mixtures and PCR set-up can also be considered depending on the outcome. One can evaluate the success of the extension PCR by comparing the bands of the extension PCR product and the fungal PCR product using gel electrophoresis.

### 3.4. Confirmation of Indexing for Illumina Library Sequencing

Index PCR confirmed that the Illumina primer pair had been indexed to the extended PCR products (called multiplexing), which track the samples at the NGS Illumina step [38–40]. Initially, we found that index PCR with extended PCR products (ITS + extension) resulted in situations with undesired bands and primer-dimers (Figure 3h, Table 2), which indicated that the extended PCR product had only been partially amplified (perhaps amplifying only fungal primer pairs). However, adjustment to index PCR with extended PCR products that targeted the short ITS2 region resulted in the desired DNA fragments, suggesting that the extension with index successfully bound with the fungal ITS2 region (Figure 3i; Table 2). Again, the success of the index PCR can be assessed by using gel electrophoresis to compare the bands corresponding to the index PCR product and extension PCR product.

### 4. Conclusions

Our study demonstrates how the selection of DNA extraction kit and PCR optimization can be used to provide insights into DNA extraction quality, primer selection, and PCR program set-ups to be carried out before HTS. Washing steps during extraction, optimization of annealing temperature and number of PCR cycles are key elements in achieving meaningful PCR setups. The amount and quality of template DNA are important, and we have shown how more samples as replicates at every step of the protocol may result in a method that can be applied to a large set of diverse environmental samples in the preparation of an amplicon library.

**Author Contributions:** The project was the result of master's thesis work by A.B.S. (Abu Bakar Siddique, abu.bakar.siddique@umu.se) supervised by B.R.A., H.I. and A.B.S. (Abu Bakar Siddique, abu.siddique@umu.se). A.B.S. (Abu Bakar Siddique, abu.bakar.siddique@umu.se) and A.B.S. (Abu Bakar Siddique, abu.siddique@umu.se) performed the molecular protocol work at Umeå Plant Science Centre (UPSC) with support from the bioinformatics facility (UPSCb). All authors have read and agreed to the published version of the manuscript.

**Funding:** Financial contributions from the Knut and Alice Wallenberg Foundation and the Swedish Governmental Agency for Innovation Systems (VINNOVA; reference numbers: VINNOVA 2016-00504; KAW 2016.0341 and KAW 2016.0352) to UPSC and UPSCb supported the study. Funding was provided by BRA supported by the UPSC Berzelii Centre for Forest Biotechnology and the Trees for the Future (T4F) project.

**Institutional Review Board Statement:** Not applicable.

**Informed Consent Statement:** Not applicable.

**Data Availability Statement:** Not applicable.

**Acknowledgments:** This work was possible due to the Erasmus Mundus Master Program in Plant Breeding (emPLANT) financed by the European Commission (Project reference: 586618-EPP-1-2017-1-FR-EPPKA1-JMD-MOB). The Kempe foundation (JCK-1919.1) supported ABS2. Bioinformatics discussions and direct access to services by SciLifeLab National Genomics Infrastructure, NGI, was provided through UPSCb.

**Conflicts of Interest:** The authors declare no conflict of interest.

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
