# Peer review of "Optimization of Protocol for Construction of Fungal ITS Amplicon Library for High-Throughput Illumina Sequencing to Study the Mycobiome of Aspen Leaves"

_applsci, doi:10.3390/app12031136_

Round 1

Reviewer 1 Report

Comments:

Introduction

Only minor revision is needed. I suggest rephrasing two sentences to make them more clear and consulting the native English speaker for the minor improvements of English language (see comments in the pdf file). The Figure 1 is very well prepared and clear and contributes to the overall clarity of the introduction.

Materials and methods

I suggest adding more details about the biological materials (leaves) collected for the research. If the leaves were received from somebody else, it is not mandatory to specify all the details I suggest in the comments in the pdf file, since this is not the topic of the paper, however, specifying as much details as possible would improve the clarity of used methodology.

Please state how many leaves were used with each DNA extraction kit.

PCR protocols (prepared master mixtures and used programs) should be explained in more details so they can be verified or repeated by other researchers.

I strongly recommend putting the figure S1 not as a supplementary material, but rather as an appendix after the main text. This way it will be easily approachable, and it doesn’t take much space.

Results and Discussion

Table one should be revised to be clearer to readers.

A sentence or two should be added, with a comment on the downsides of using the NanoDrop for the DNA quantification. NanoDrop is usually used for the quantification of DNA, however, it is not as precise as some other devices, such as fluorometers (Qubit, Quantus and similar). This does not mean that the results are not useful or should be discarded, on the contrary.

Optimized PCR conditions should be clearly stated in detail.

The last part of the section titled Confirmation of indexing for Illumina library should be moved to Materials and methods.

English language requires moderate revisions made by a native speaker throughout the entire text.

Author Response

Thank you for allowing us to resubmit our paper with minor changes. Attached you will find our point-by-point rebuttal to the reviewers' comments.

Reviewer 1

Introduction. Only minor revision is needed. I suggest rephrasing two sentences to make them more clear and consulting the native English speaker for the minor improvements of English language (see comments in the pdf file). The Figure 1 is very well prepared and clear and contributes to the overall clarity of the introduction.

Answer: We have edited the manuscript after suggestions and used services by Sees editing to improve the language.

Materials and methods. I suggest adding more details about the biological materials (leaves) collected for the research. If the leaves were received from somebody else, it is not mandatory to specify all the details I suggest in the comments in the pdf file, since this is not the topic of the paper, however, specifying as much details as possible would improve the clarity of used methodology.

Answer: we were responsible for the biological material and have included a detailed description of its origin and handling. Lines: 76-92.

Please state how many leaves were used with each DNA extraction kit.

Answer: we harvested ten leaves per plant. Line 78.

PCR protocols (prepared master mixtures and used programs) should be explained in more details so they can be verified or repeated by other researchers.

Answer: We have added more details to the PCR protocol and specified concentrations of reagents etc to make the protocol details more assessable to readers. Lines: 108-148.

I strongly recommend putting the figure S1 not as a supplementary material, but rather as an appendix after the main text. This way it will be easily approachable, and it doesn’t take much space.

Answer: We thank you for this option and have included now renamed figure s1, and included it as figure 3 and accordingly changed references in the manuscript.

Results and Discussion

Table one should be revised to be clearer to readers.

Answer: Effort has been made to revise Table one. Particularly we added explanations to column content and abbreviations.

A sentence or two should be added, with a comment on the downsides of using the NanoDrop for the DNA quantification. NanoDrop is usually used for the quantification of DNA, however, it is not as precise as some other devices, such as fluorometers (Qubit, Quantus and similar). This does not mean that the results are not useful or should be discarded, on the contrary.

Answer: We have extended the discussion on the use of Nanodrop for DNA quality assessment and discussed the use of alternative measurements. Lines:172-178.

Optimized PCR conditions should be clearly stated in detail.

Answer: Optimized PCR conditions have been detailed in line: 108-140 and in table 2's footnote.

The last part of the section titled Confirmation of indexing for Illumina library should be moved to Materials and methods.

Answer: We followed this suggestion and moved the text to the material and method section.  Lines 141-149.

The English language requires moderate revisions made by a native speaker throughout the entire text. Answer:  We used the services provided by Sees editing to improve the language.

Reviewer 2.

Line 41: Consider rephrasing this part of sentence, since the similar phrase is already used in the first part. I believe this would make the sentence more clear to a reader.

Line 54: Please consider rephrasing this sentence, especially this last part, to make it more clear. Perhaps:....protocols, which were optimized for... I believe this can be written without capital P: polymerase. Answers: These text parts have been rephrased.

Figure 1: Excellent figure! Very well prepared and clear! Answer: Thanks.

Materials and Methods:

2.1 Biological samples: Please specify were the leaves collected from one or more trees, how many if more, and if known, please also specify at least the approximate age of these trees, as these are all factors which can affect the mycobiota of leaves..... Please specify in which period(s) of the years; spring, summer, or early autumn?

Answer: Please find a detailed description of the origin and handling of the biological material. Lines: 76-92.

Line 66: I recommend naming the exact lab were the preparation of leaves was conducted.

Answer: The UPSC environment was used to prepare the samples and the by SciLifeLab National Genomics Infrastructure, NGI used for the sequencing, which is now detailed in the materials section. Lines 76-92.

2.2. DNA extraction:  Line #72: It would be clearer to combine this in one sentence.

Answer: Thank you this text is reworded.  Lines 95-98.

2.3 Primers, PCR and gel electrophoresis: Line 82: You don't need to emphasize that this was fungal PCR, since fungal DNA was the target in all PCRs. You should make a clear distinction between the PCR with specific fungal primers and the one with extended primers.

Answer: We have restructured this paragraph and divided it into three parts to illustrate the steps described in Figure 1. Lines 108-140.

Line 85: I assume two different volumes were used for two different PCRs, one for the PCR with fungal specific primers, and other for the PCR with extended primers? You should somehow make this more clear, maybe you can write a sentence clarifying this before stating the used amounts/concentrations of the ingredients in the PCR master mix.

Answer: We have added exact volumes and concentrations in the revised text.

Line 87: You should by all means specify the program conditions, exact time and temperatures of each step, for both PCRs, of course, with the exception of conditions which were modified/optimized.

Answer: We have added these details for the three kinds of primers and refer to steps during the optimization process with references to table 2. Mainly lines 108-140.

Line 98: Which dye was used for the visualization?

Answer: We used GelRed. Line 114.

  1. Results and Discussion

Line 101: Discussion, not with s

3.1 Choice of extraction kit...Line 102: the Extraction Kit (but please verify with the native speaker) Answers: the manuscript was language checked by the Sees editorial service

Line 111: Psifidi et al., line 111: Number of the citation should be put in bracket, for example

Psifidi et al. [13] Answer: This has been taken care of, throughout.

Line 99: In this section of the Results and Discussion you should add a sentence or two where you should critically comment your own methodology. NanoDrop is usually used for the quantification of DNA, however, it is not as precise as some other devices, such as fluorometers (Qubit, Quantus and similar).

Answer: We have included this issue. Lines 172-178.

Table 1. It would be convenient to readers if you put the optimal expected values of 260/280 and 260/230 ratios in the table footnote for example.

Answer: We have added information to Table 1's footnote including an indication of the range of accepted values for 260/280 and 260/230 ratios.

3.2 Fungal primer pair optimization

Line 127: Please rephrase this sentence, it is not clear.

Answer: this sentence now reads: “The five fungal specific primers (Figure 1a, point 4) were tested according to the initial protocol settings above and optimization adjustments were performed mainly by the choice of fungal ribosomal primers specific to the ITS region they targeted. Lines: 116-118.

Line 141, Figure 2:, Table 2.: You should definitely specify how exactly and to which values did you modify these conditions. In table 2, What exactly were the conditions if the settings are labeled 1.56/30? Answer: we have now detailed the conditions throughout the manuscript.  1. 56/30 means ‘1’ is a serial number of that event, ‘56’ is the annealing temperature and ‘30’ is the number of cycles. We added the explanation as a footnote to table 2.

3.3 Optimization of Extension PCR

Line 156: However, the bands.... instead of But, Answer: the text is rephrased

Line 161: Rephrase. Answer: the text is rephrased

Line 168: Rephrase. Answer: the text is rephrased

3.4 Confirmation of indexing for Illumina library

Line 183: you should finish the sentence with the appropriate verb Answer: the text is rephrased

Reviewer 2 Report

The manuscript (Protocol Optimization of Fungal ITS Amplicon Library for High-Through- put Illumina Sequencing to Study the Mycobiome of Aspen Leaves) is interesting and introduce an optimized protocol to prepare the amplicon library for the intended samples for HTS.

The English should be carefully checked again with a native English speaker.

I think you should write more about the impact of your study and your results with examples. The aims were clearly explained but still, you have to write more in different parts of the manuscript about the applications of your study’s results.

In my opinion, in the introduction, you should add some examples of the utilization of HTS. You described the steps and you should follow them with different examples and studies.

L66. And L156. Please correct the symbol of Celsius.

  1. Please add a comma after extraction.

L112. Please correct to chose to preferred.    

L125. Please add fungal ribosomal specific primers before (Ps).

L198. The sentence (showed how replication at every step of the protocol) is not clear, rewrite please.

Author Response

Reviewer 2 The manuscript (Protocol Optimization of Fungal ITS Amplicon Library for High-Through- put Illumina Sequencing to Study the Mycobiome of Aspen Leaves) is interesting and introduce an optimized protocol to prepare the amplicon library for the intended samples for HTS. I think you should write more about the impact of your study and your results with examples. The aims were clearly explained but still, you have to write more in different parts of the manuscript about the applications of your study’s results.

Answer: Application of results are now treated in lines 161-164; 172-178; 184-187; 202-204; 231-237; and 241-257.

In my opinion, in the introduction, you should add some examples of the utilization of HTS. You described the steps and you should follow them with different examples and studies.

Answer: The introduction has been rewritten and arguments added for HTS-studies. Lines 42-53.

L66. And L156. Please correct the symbol of Celsius. Answer: Celsius symbol has been corrected throughout the manuscript.

L104. Please add a comma after extraction. Answer: corrected.

L112. Please correct to chose to preferred.   Answer: corrected.

L125. Please add fungal ribosomal specific primers before (Ps). Answer: added

L198. The sentence (showed how replication at every step of the protocol) is not clear, rewrite please. Answer: The sentence is reworded.